# Bronchial Asthma as a Cardiovascular Risk Factor: A Prospective Observational Study

**DOI:** 10.3390/biomedicines10102614

**Published:** 2022-10-18

**Authors:** Marcela Kreslová, Olga Kirchnerová, Daniel Rajdl, Vendula Sudová, Jiří Blažek, Aneta Sýkorová, Petr Jehlička, Ladislav Trefil, Jan Schwarz, Renata Pomahačová, Josef Sýkora

**Affiliations:** 1Department of Paediatrics, Faculty Hospital, Faculty of Medicine in Pilsen, Charles University in Prague, Alej Svobody 80, 30460 Pilsen, Czech Republic; 2Department of Pneumology, Faculty Hospital, Faculty of Medicine in Pilsen, Charles University in Prague, E. Beneše 1128, 30100 Pilsen, Czech Republic; 3Department of Clinical Biochemistry and Haematology, Faculty Hospital, Faculty of Medicine in Pilsen, Charles University in Prague, Alej Svobody 80, 30460 Pilsen, Czech Republic

**Keywords:** endothelial dysfunction, asthma, reactive hyperemia index, high-sensitive CRP, biomarkers, atherosclerosis

## Abstract

Introduction: Asthma as a chronic inflammatory disorder has been suggested as a risk factor for endothelial dysfunction (ED), but studies on the association between asthma and cardiovascular disease (CVD) risk are limited. Background: We assessed associations of ED with the severity of asthma, eosinophilic inflammation, lung function, and asthma control. Methods: 52 young asthmatics (median age of 25.22 years) and 45 healthy individuals were included. Demographic, clinical, and laboratory findings were recorded. We evaluated microvascular responsiveness by recording the reactive hyperemia index (RHI) indicating post-occlusive peripheral endothelium-dependent changes in vascular tone using the Itamar Medical EndoPAT2000. VCAM-1, ADMA, high-sensitive CRP (hsCRP), and E-selectin were measured. Results: Asthmatics had considerably lower RHI values (*p* < 0.001) with a dynamic decreasing trend by asthma severity and higher hsCRP levels (*p* < 0.001). A substantial increase in hsCRP and E-selectin with asthma severity (*p* < 0.05) was also observed. We confirmed a higher body mass index (BMI) in asthmatics (*p* < 0.001), especially in women and in severe asthma. Conclusions: We demonstrated the progression of CVD in asthmatics and the association of the ongoing deterioration of ED with the inflammatory severity, suggesting that the increased risk of CVD in young asthmatics is dependent on disease severity. The underlying mechanisms of risk factors for CVD and disease control require further study.

## 1. Introduction

Bronchial asthma is characterized by chronic bronchial inflammation of variable intensity accompanied by recurrent reversible airflow obstruction and symptoms such as coughing, wheezing, shortness of breath, and chest tightness [1], with a worldwide increase over the last several decades.

Accumulating evidence suggests that low-grade chronic systemic inflammation and autoimmune diseases are risk factors for cardiovascular disease (CVD). Several studies have shown an increased risk of CVD in patients with asthma [2,3,4,5], chronic obstructive pulmonary disease (COPD) [6,7,8,9,10], cystic fibrosis (CF) [11,12,13,14], alpha1-antitrypsin deficiency [15,16], and obstructive sleep apnea [17]. For improved prognosis and to prevent amplifying CVD, understanding the association between pulmonary diseases and CVD has important implications for disease management and targeted treatments. In addition, it is still largely unknown which potential biological mechanisms of premature manifestation of CVD, known as subclinical atherosclerosis (AS), are involved in current and future cardiovascular risk in young asthmatics. On the other hand, evidence has shown that chronic systemic inflammation plays a key role in the pathogenesis of endothelial dysfunction (ED) and asthma: asthma symptoms are caused by inflammation and narrowing of the peripheral airways. Asthma, as well as COPD, may be associated with increased inflammation and increased CVD events [18]; severe asthma is especially associated with massively increased mucus production and neutrophilic airway inflammation.

ED is a systemic disorder that represents a key variable in the development of AS and its complications. AS is characterized by lipid deposition and chronic inflammation, associated with immune activation and the induction of inflammatory mediators and signaling pathways [19]. Most commonly, impaired endothelium-dependent vasodilation is presented as a hallmark of ED. In the earliest stages, there is only functional damage of the endothelial cells due to mechanical, physicochemical, and immunological factors. ED is characterized by a reduced bioavailability of vasodilators—in particular, nitric oxide (NO)—to a suitable ischemic stimulus, a procoagulant and proinflammatory state that leads to a proatherosclerotic structure, whereas endothelium-derived contracting factors are increased [20]. Proangiogenic activity and increased endothelial growth factors in asthma patients have been confirmed [21,22], similar to chronic bronchitis patients. Vascular endothelial growth factor (VEGF) is a highly specific growth factor for endothelial cells that is produced in response to hypoxia and induces cell proliferation and migration and prevents endothelial cell apoptosis [23]. At present, it seems that NO released in the lower respiratory tract, measured as fractional exhaled nitric oxide (FENO), plays a crucial role in the pathophysiology of asthma and acts as an indirect biomarker of eosinophilic airway inflammation. Furthermore, an increased value of ECP in the peripheral blood and eosinophilia in the differential budget of the blood count can also indicate an endobronchial eosinophilia. FENO identifies eosinophilic airway inflammation, the prognosis of exacerbations, a decline in lung function, and the degree of airway hyperreactivity. FENO measurement helps in predicting the response to treatment with corticosteroids and biologics and assessing treatment adherence and treatment success [24].

Despite such evidence, there can be little doubt that the reversibility of ED and the availability of early detection of ED may have therapeutic and prognostic implications for reducing CVD morbidity and mortality [25]. Asthma is a severe chronic inflammatory disorder with high associated morbidity, as well as COPD. However, the evidence for ED in asthma is not as great as that in COPD [7]. ED is typically present in asthmatic adults [26]. Reduced flow-mediated dilatation (FMD) levels in asthmatics—a relation of FMD to asthma severity confirming a worsening endothelial function with disease progression—are also found in patients with COPD [27]. NO synthase gene polymorphism and angiotensin-converting enzyme as key elements of asthma in ED and vascular aging have been highlighted, and genetic predisposition to ED in asthma has been examined by Cortez e Castro et al. [28]. As far as we know, our knowledge of ED and higher CVD risk and the biological mechanisms in pediatric and young asthmatic patients is incomplete.

## 2. Aims

With this in mind, we decided to illuminate CVD risk in young patients with asthma using a combined diagnostic approach by measuring RHI and specific vascular biomarkers of ED. The aims of the study were to compare RHI in patients with asthma and healthy individuals (HI) and to ascertain the associations between RHI and biomarkers. Furthermore, the interrelationship of endothelial function parameters and CVD risk factors, such as pulmonary function, endobronchial eosinophilic inflammation, and the control and severity of the disease, was also examined.

## 3. Materials and Methods

### 3.1. Study Population

This prospective cohort study was performed at the Pneumology Unit of the Departments of Pediatrics and the Special Unit for the Treatment of Difficult-to-Treat Asthma, and the Departments of Pneumology, Charles University in Prague, Faculty Hospital Pilsen. During the study period, a total of 52 consecutive subjects over 14 years of age with bronchial asthma of varying severity on regular anti-asthmatic treatment, without signs of acute exacerbation of asthma, otherwise without known vascular disease, were enrolled. Asthmatics regularly received treatment with inhaled corticosteroids (ICS), inhaled corticosteroids plus long-acting beta-agonists (ICS/LABA), inhaled corticosteroids plus long-acting beta-agonists plus long-acting muscarinic antagonists (ICS/LABA+LAMA), leukotriene receptor antagonists (LTRA), theophylline, oral corticosteroids (OCS), the anti-immunoglobulin E (IgE) monoclonal antibody omalizumab (OMA), or a combination of these agents. Nutritional status was assessed using BMI, endobronchial eosinophilic inflammation was assessed by measuring fractional exhaled nitric oxide (FENO) using the NIOX VERO^®^ device, and lung function was evaluated by spirometric forced expiratory flow between 25 and 50% of forced vital capacity (FEF_25–50_) using the Flowhandy ZAN 100^®^ (nSpire Health^®^, Inc., Oberthulba, Germany). Asthma control was assessed using the Asthma Control Test (ACT).

The comparison group consisted of 45 healthy subjects with no history of heart, metabolic, inflammatory, neoplastic, or peripheral vascular disease, as well as with no antibiotic, anti-inflammatory, and vasoactive treatment affecting endothelial function, or any chronic disease known to affect microvascular function. Exclusion criteria for both groups were dyslipidemia, smoking, positive family history of premature cardiovascular events, and abnormal left-ventricular function. The non-inclusion of smokers and obese subjects in the comparison cohort is consistent with evidence of increased baseline pulse amplitude in obese people with metabolic syndrome and an inverse relationship between baseline amplitude and peripheral arterial tone (PAT) response to hyperemia [20].

Healthy individuals were recruited in the same period from the same community, thereby avoiding problems of bias resulting from inappropriate controls. All patients were treated in a single central unit in a uniform manner.

### 3.2. Reactive Hyperemia Index Measurements

RHI was detected by non-invasive plethysmographic examination using the Endo-PAT^TM^2000 device (Itamar Medical Ltd., Caesarea, Israel) based on the evaluation of endothelial-dependent changes in post-occlusive PAT sensed by biosensors placed on the index fingers. The measurements were performed on the examinees in a supine position, in a quiet, thermoneutral room. Initially, body mass index (BMI) and blood pressure on the non-dominant contralateral arm were measured. Input PAT values on the upper limbs were recorded during the first phase and followed by a five-minute occlusion of the brachial artery on the non-dominant limb with a pressure cuff inflated 60 mm Hg above the systolic pressure, at least 200 mm Hg. The post-occlusive endothelium-induced dilation with reactive hyperemia was captured as an increase in the PAT signal amplitude. The EndoPAT index (RHI) was automatically calculated from the ratio of the occlusive and pre-occlusive arterial flow relative to the values of the simultaneously measured, non-closed contralateral limbs. An insufficient increase in PAT amplitude during the last post-occlusive phase was associated with ED. The RHI cutoff was set at 1.67 in adults. A value less than 1.67 is considered pathological; a score above 2.10 is recommended [29,30].

### 3.3. Laboratory Analysis

We evaluated four biomarkers related to ED to determine the subclinical stage of AS: vascular cell adhesion molecules E-selectin and VCAM-1 and a competitive NO-synthase inhibitor dimethylarginine (ADMA), all measured by ELISA method (DLD Diagnostika, Hamburg, Germany), and a high-sensitive C-reactive protein (hsCRP) measured by a particle-enhanced immuno-turbidimetric assay technique (Aidian, formerly Orion Diagnostica, Espoo, Finland). After overnight fasting, blood samples were obtained from all subjects in a standard clinical setting and further examined in a blinded fashion. Serum obtained from blood samples was frozen within 2 h after blood collection. All markers were determined batchwise. In asthmatics, we evaluated the level of total IgE and allergen-specific IgE (sIgE), eosinophil cationic protein (ECP) in peripheral blood, and eosinophilia in the differential budget of the blood count.

### 3.4. Statistical Analysis

The statistical analysis was performed using SAS 9.4 software (SAS Institute^®^, Cary, NC, USA). Values obtained from individual measurements were expressed using descriptive statistics. Statistical differences were calculated using a non-parametric Wilcoxon two-sample test or its generalized variant, the Kruskal-Wallis test. Correlations between RHI and specific biomarkers and the relation of RHI to CVD risk factors were assessed using Spearman’s and Kendall’s rank correlation. For all analyses, a *p*-value ≤ 0.05 was considered statistically significant.

### 3.5. Ethics

The study was conducted in accordance with the Declaration of Helsinki, and the protocol was approved by the local Ethics Committee of the Faculty Hospital in Pilsen (reference 129/20). All subjects or their legal representatives gave their informed consent for inclusion and data processing before they participated in the study.

## 4. Results

### 4.1. Demographic Data of the Study Population 

The characteristics of the study subjects are shown in Table 1. Group distribution according to selected criteria (type and severity of asthma, eosinophilic airway inflammation assessed by FENO, lung function evaluated by spirometry FEF_25–50_, asthma control) is shown in Table 2. HI corresponded to the group of asthma patients in terms of the number and representation of sex and ethnicity. The representation of asthmatics by the type of treatment in relation to asthma severity is shown in Table 3. The examined asthmatics belonged to the T2-high eosinophilic-allergic phenotype, except a small specific group (six patients) of exercise-induced asthma [31].

In our cohort, nine patients (17.3%) fulfilled the diagnostic criteria of severe refractory bronchial asthma by treatment with a humanized recombinant monoclonal anti-IgE antibody. They belonged to the T2-high allergic phenotype with eosinophilia. The adherence of patients to omalizumab treatment was excellent; the initial ACT before the start of anti-IgE treatment was on average 10–13 points, and after treatment there was an improvement to the values of 17–19 points. No patients were taking any other biologic drugs.

Thirty-six asthmatics (69.2%) received inhaled anti-asthmatic therapy without biological therapy. Seven patients (13.5%) were on LTRA therapy without inhaled anti-asthmatic therapy. They belonged to the mild allergic asthma group (T2-high phenotype) and had full asthma control. Patients in the exercise-induced asthma group were treated with ICS or ICS/LABA.

Six patients (11.5% of examined asthmatics) had severe or very severe obstruction of peripheral airways (FEF_25–50_ 13–38%), with good adherence to treatment regardless of gender (three men, three women). These patients mainly belonged to the severe asthma group (five severe asthmatics on anti-IgE therapy and one moderate asthmatic on ICS/LABA + antiHIS treatment).

According to the current FENO level [24], 21.2% of patients had a positive finding of florid eosinophilic bronchitis, and 46.2% of examined patients achieved full disease control.

The level of total IgE, ECP, blood eosinophils count in the differential budget of the blood count and the positivity of sIgE are shown in Table 4. Regarding eosinophilia, in our group of patients with severe asthma, 50% were negative and 50% positive, which may be due to the simultaneous use of systemic corticoids distorting these results (blood eosinophil count, IgE, ECP).

In asthmatics, there was a statistically significantly higher diastolic arterial blood pressure (*p* < 0.005) as well as a statistically significantly higher BMI compared to HI (*p* = 0.0002, Figure 1). In terms of gender, a higher BMI was confirmed in women (24.73 vs. 23.58). In terms of asthma severity, a higher BMI was confirmed in patients with severe and severe refractory asthma (28.60 vs. 23.20). 

### 4.2. RHI and Biomarkers

RHI values were statistically significantly lower in asthmatics (1.53 vs. 1.81; *p* < 0.001), as shown in Table 5 and Figure 2. A statistically significant difference between asthmatics and HI (*p* = 0.001), with a specificity of 72.73% and a sensitivity of 61.54%, was found at the RHI cutoff point of 1.67.

The results of the determined biomarkers are summarized in Table 5. A statistically significantly higher level of hsCRP was demonstrated in asthma patients (0.9 vs. 0.25; *p* < 0.001), as shown in Figure 3. Statistically significantly lower levels were found for ADMA and VCAM-1 markers. E-selectin levels between asthma patients and HI did not reach statistical differences. 

### 4.3. Correlation Analysis

In asthma patients, no correlation of RHI with biomarkers, asthma control, or nutritional status was found. A moderate correlation of BMI with hsCRP (r = 0.461; *p* < 0.001) and a low negative correlation of BMI with VCAM-1 (r = −0.312; *p* < 0.05) were confirmed. A low negative correlation of RHI with ADMA was demonstrated in HI (r = −0.407; *p* < 0.05).

A dynamic trend of decreasing RHI with allergic asthma severity was observed (median 1.52 mild asthma vs. 1.50 moderate asthma vs. 1.48 severe and severe refractory asthma; *p* = 0.905). The median RHI for exercise-induced asthma was 1.67, which may be related to a lower CV risk, given that these patients had full asthma control. Comparing the most severe asthmatics (severe and severe refractory asthma) with others, we measured a lower RHI without proven statistical significance (median 1.48 severe asthma vs. 1.54 other asthma patients; *p* = 0.620).

A dynamic trend of decreasing RHI with allergic asthma inflammatory severity evaluated by FENO was indicated (median 1.60 normal FENO vs. 1.52 mid-range FENO vs. 1.42 positive values; *p* = 0.736) as well as a lower RHI median for the FENO positive group compared to other patients (1.42 FENO positive vs. 1.55 others; *p* = 0.477), both without proven statistical significance. 

Regarding lung function, a difference in RHI was indicated in the group of patients with severe and very severe obstruction compared to other patients (median 1.43 vs. 1.54 others) without proven statistical significance (*p* = 0.909). 

Biomarker results showed a dynamic trend of ADMA levels decreasing with allergic asthma severity (median 0.48 mild asthma vs. 0.46 moderate asthma vs. 0.41 severe and severe refractory; *p* < 0.05). ADMA levels were significantly lower in asthma patients compared to HI (*p* < 0.05). A statistically significant difference was demonstrated in hsCRP according to the type of asthma (median 3.35 severe and severe refractory asthma vs. 0.44 other asthmatics; *p* = 0.012). There was a statistically significant difference in E-selectin by asthma severity (median 39.95 severe and severe refractory asthma vs. 58.20 other asthmatics; *p* < 0.05). In VCAM-1, a decreasing trend with the severity of allergic asthma was indicated (median 851.00 mild asthma vs. 765.70 moderate asthma vs. 698.15 severe and severe refractory asthma; *p* = 0.390). 

Regarding FENO, a lower median of VCAM-1 in the FENO positive group compared to other patients was found (708.70 FENO positive vs. 773.20 others; *p* = 0.05). 

Regarding biomarkers and disease control, a decreasing trend with poor asthma control was indicated for markers VCAM-1 (median 851.25 full control vs. 711.35 insufficient control; *p* = 0.133) and ADMA (median 0.46 full control vs. 0.40 insufficient control; *p* = 0.143). For E-selectin, a statistically significant difference was demonstrated (median 59.0 full control vs. 37.2 insufficient control; *p* < 0.01). 

In relation to lung function, an increasing trend was found for hsCRP with obstruction severity (median hsCRP 0.41 normal lung function vs. 1.30 mild obstruction vs. 1.75 moderate obstruction vs. 2.85 severe and very severe obstruction, without proven statistical significance (*p* = 0.128)).

## 5. Discussion

To the best of our knowledge and research, this is the first published clinical report ever undertaken to explore ED in young asthmatic patients using a combination of RHI and biomarkers. Our findings support the hypothesis of increased CVD risk in young asthmatics and offer new insights into CVD in asthmatic patients. The results of our study confirm the negative effect of asthma on endothelial function. The other main findings can be summarized: First, we found significantly different ED values in asthma patients compared with the normal response in HI. Second, the finding of an attenuated post-occlusion hyperemic microvascular response may suggest a possible association with premature ED in severe asthma patients, confirming the association of the development of ED with the severity of asthma and disease progression. Third, there is ample evidence to suggest that differences in ED, as seen in our targeted group of asthmatics, implicate some pathophysiological mechanisms responsible for CVD progression in asthma. Although the pathway analysis requires further study, the observed patterns may facilitate focused research on the pathological processes responsible for ED in this group of patients. We anticipate that these results will substantiate the need for targeted care for high-risk asthma patients. Thus, these findings and the knowledge of ED in asthma may have important implications for therapeutic strategies, as severe refractory asthma is a serious problem with high morbidity and mortality in patients and a significant economic burden on healthcare [32]. Moreover, Makieieva et al. reported the dependence of asthma severity on the functional state of the vascular endothelium [3].

Plenty of records concerning the effects of ED have indicated that the status of an individual case of ED may reflect the propensity to develop AS and may serve as a marker of an unfavorable CVD prognosis [33]. In most young patients with asthma, there are no symptoms of CVD complications. This creates the need for introducing minimally invasive measures that would allow the early detection of ED. However, the early detection of microvascular dysfunction seems to be particularly difficult. Attempts to detect vascular disease early are associated with evidence of morphological changes as early as childhood [20] when the clinical course is still quite inconspicuous and reversible. Bronchial asthma is associated with increased arterial inflammation beyond that estimated by current risk stratification tools. Several studies report a link between chronic inflammatory diseases of the respiratory tract activating systemic inflammatory biomarkers and increased incidence of CVD [4,5,34]. These observations suggest that chronic inflammatory challenges may increase arterial stiffness, a risk factor that may have contributed to CVD, which increases with disease severity [35,36].

Although asthma is the most common chronic disease in children, unlike many studies in adults there is insufficient evidence of ED in children. As the quality of life of asthmatics improves, asthma can be expected to be co-morbid in adults, accompanied by a chronic systemic inflammatory process combined with a prothrombotic status and oxidative stress. Several plausible mechanisms have been suggested to explain an increased risk of CVD among asthmatic subjects and the underlying mechanisms involved. One possible explanation for these observations could be that they indicate a new understanding of the mechanisms involved in the development of major proatherogenic CVD risk factors, including the systemic inflammatory process and oxidative stress, prothrombogenic factors, dyslipidemia, high fat intake, and low physical activity [23]. Previous studies have also suggested that oxidative stress promoting cellular damage is the main pathophysiologic mechanism resulting in a reduced NO bioavailability and in the facilitation of ED, as the vascular endothelium is a major target for reactive oxygen species by reacting with NO [23]. The airways of obese asthmatics are deficient in NO, leading to a reduced response to ICS. L-citrulline, a precursor of L-arginine recycling and NO generation, can reduce oxidative stress and improve asthma [37]. In addition, lung function impairment could play a role in the association of asthma with CVD risk.

In the current sample, the most important findings of our study are reduced RHI values in asthmatics (1.53 vs. 1.81; *p* < 0.001), a decreasing trend of ADMA (*p* < 0.05) and E-selectin levels (*p* < 0.05), increasing hsCRP levels (*p* < 0.05) with asthma severity, a statistically significant difference in E-selectin in relation to asthma control (*p* < 0.01), and decreasing VCAM 1 levels with the severity of eosinophilic airway inflammation (*p* = 0.05). It should be noted, however, that although elevated levels of the biomarkers VCAM-1 and E-selectin were demonstrated in our previous cohort of patients with CF [38], there appeared to be no significant increase in these biomarkers in asthmatics in the current trial. Some researchers have reached controversial conclusions. Our result for asthmatic patients did not coincide with that of Hamzaoui and colleagues who reported an elevation in serum-soluble E-selectin and VCAM-1 levels in women with severe asthma [39]. This difference could be explained by a different type of severe asthma treatment (oral glucocorticoids and theophylline) and the older age and the gender (women only) of the patients. Although the aforementioned results seem encouraging, more studies are required to understand the relationship among vascular biomarkers, inflammatory markers, and ED response in young asthmatics. The immunopathology of asthma is complex, and it can be expected that the combination of multiple biomarkers may provide more useful prognostic information than that provided by individual biomarkers.

Another concern of our analysis was hsCRP. Many studies have recognized that hsCRP is associated with an atherogenic process [40]. It has been suggested that, from a purely clinical standpoint, elevated levels are a predictive factor of morbidity and mortality in CVD events even in healthy individuals, regardless of commonly known risk factors [41]. In this respect, it is well established that an elevated baseline hsCRP may reflect a higher inflammatory load. Our data suggest more explanations for the hsCRP levels, which may be used as a surrogate marker of asthma control, with lower levels indicating better disease control. Thus, an increased hsCRP level in asthmatics suggests the importance of the chronic systemic inflammatory process as a significant risk factor, where the progression of changes in asthma is greatly affected by the chronic inflammation if not controlled adequately, which may indicate the importance of the early initiation of inhaled preventive anti-inflammatory therapy in childhood to suppress inflammatory processes in the airways. Corticosteroids broadly reduce type 2 inflammation [42]. Their anti-inflammatory effect was also confirmed by the suppression of epithelial VCAM-1 expression by Atsuta et al. [43]. Furthermore, looking at the results of our previous research, we recently reported significantly increased levels of hsCRP in CF patients since childhood as a sign of endothelial proinflammatory activity associated with a chronic systemic inflammatory process [38,44], consistent with our reported data on children with Crohn’s disease [45] and ALL [46]. Moreover, another study demonstrated the positive effect of ICS from the reduction of vascularity in bronchial biopsy specimens [47] and from the reduction of VEGF levels after six months of ICS treatment [48]. These findings suggest a key role of inflammation in increased vascularity and vascular remodeling in asthma. ICS are a key treatment for controlling asthma and preventing asthma attacks [49,50]. However, the response to ICS varies among individuals. MicroRNAs can predict the response to ICS asthma treatment over time [49]. ICS can also reduce sVCAM levels [51]. Later, Wanner and Mendes also confirmed the corticosteroid effect on endothelium-dependent vasodilation regeneration [52]. Further, LTRA may reduce the incidence of CVD events [53]. In recent years, a strong correlation between neovascularization and asthma severity has been confirmed, as well as the impact of profibrotic mediators and ED markers on pulmonary function in patients with uncontrolled moderate asthma [54]. Impaired lung function is one of the main risk factors for the premature manifestation of CVD [55]. Further preclinical and clinical studies would indicate the extent to which anti-asthmatic drugs and controller medications may have therapeutic potential in CVD.

A second major concern of our study in a cohort of asthmatic patients was a decreasing trend of RHI and increasing hsCRP levels with the severity of obstruction. Regarding vascular laboratory biomarkers, in the current study we assumed that hsCRP and E-selectin can be considered as indicators of asthma severity. ADMA marker as a competitive inhibitor of NO synthase significantly reduces NO effect. However, in the present study we have not confirmed ADMA as an ED-related mediator. Thus, our results underline the importance of investigating the NO synthase gene polymorphism in relation to the mechanisms that are active in the development of ED in these patients, as previously discussed in this same line of research by Cortez e Castro et al. [33]. Genetic biomarkers have the advantage of not changing with time; thus, it is likely that there are certain genetic polymorphisms which may be of greater utility when applied as part of a predictive model with clinical and other predictive factors in young asthmatics and may suggest targets for treatment strategies. A higher peripheral blood eosinophil count and a wider diurnal variation in peak expiratory flow would be predictive markers of unstable asthma after ICS reduction [56]. Increasing interest in the microbiome of the upper airway, lower airway, and gastrointestinal tract, and its relationship to asthma pathogenesis is helping to uncover new therapeutic targets. There is evidence that the airway microbiome may even predict the response to ICS [57].

Up to now, evidence supporting the gender effect in asthmatics has been obtained in only a few observational studies [4,5,58]. Viewed by gender, the results of our present study revealed a tendency toward higher CVD risk in women with asthma, which is consistent with other inflammatory immune diseases, such as IBD, rheumatoid arthritis due to estrogen influence, and generally higher BMI [59,60]. Furthermore, it is also apparent from our study that BMI correlates with hsCRP (r = 0.461; *p* < 0.001). Finally, according to previous reports, it was clarified that a lack of physical activity in patients with severe allergic asthma and the influence of diet and age can also be considered as possible causes [19,61]. 

Various mechanisms play a role in the etiology of asthma, including allergies. The allergic asthma inflammatory severity evaluated by FENO is associated with risk markers of AS and with diseases that secondarily lead to the progression of AS. It should be stressed that a dynamic trend of decreasing RHI with increasing FENO levels is evident in our study. An inverse correlation of FENO in patients with ischemic heart disease to some risk markers of AS such as plasma levels of triglycerides and glycated hemoglobin was demonstrated, most likely on the basis of ED with consequent decreased NO production and increased NO degradation in hyperglycemia and higher triglyceride concentrations [62]. The endothelial importance in the development and exacerbation of allergic diseases, corticosteroid-refractory reactions involved in the refractory processes of allergic disorders, and the use of endothelial-targeted therapy as a treatment option for corticosteroid-resistant allergic disorders were confirmed [63]. A reduction in FENO in asthmatic children on a maintenance dose of ICS treated with LTRA evidences the anti-inflammatory effect of montelukast additive to the effect of ICS. [64]. Yin SS et al. also investigated the correlation between FENO and ICS efficacy in childhood bronchial asthma [65]. Syk et al. highlights the possibility of IgE reduction with common asthma control drugs, which is confirmed in patients with persistent atopic asthma with annual optimization of ICS and LTRA treatment, resulting in a significant decrease in total IgE correlating with a reduction in FENO and an improvement in asthma control and quality of life [66]. Total IgE may predict asthma severity and the risk of exacerbations and loss of asthma control. The combination of FENO with lung function indices had predictive value in asthma response. Elevated FENO is a clinical indicator of uncontrolled asthma in children receiving ICS. Blood eosinophil counts are used to define eosinophil phenotype, predict the response to biologic therapy, and predict exacerbations, loss of asthma control, and decline in lung function. Regarding eosinophilia, in our group of severe asthmatics there were 50% negative and 50% positive patients, perhaps due to the use of systemic corticoids, which can distort the levels of eosinophilia and allergy.

Antiselectin therapy by inhibiting leukocyte migration to target organs appears promising for the treatment of inflammatory diseases including asthma [67]. Soluble E-selectin could be considered an objective marker of inflammation and the severity of atopic dermatitis in children [68]. Moreover, selenium can affect the adhesion molecule expressions in corticoid-dependent asthmatics, such as E-selectin and VCAM-1, which are crucial in the chronic inflammatory process [69]. High serum IgE levels and evidence of atopy are the biomarkers presently used for the selection of patients for omalizumab treatment. 

In general, asthma is associated with subsequent increased risks of cardiovascular heart disease (atrial fibrillation, heart failure, and myocardial infarction) and cardiovascular mortality. This relationship was confirmed by Hua et al. as well as a higher risk of CVD in females with asthma, while active asthmatic patients were found to be at a higher risk of cardiovascular mortality than non-active asthmatic patients. These data have suggested the necessity of early detection and intervention in asthma patients [58]. Several studies evaluated the effect of bronchodilators, LAMA, and theophylline in relation to CVD [70,71].

Adherence to medication is crucial in patients with severe asthma. High adherence to omalizumab is associated with better outcomes and control of asthma [72]. Likewise, mepolizumab reduces corticosteroid intake and asthma symptoms in patients with severe asthma even in the subgroup with coexisting bronchiectasis, independently of their severity [73]. Omalizumab can decrease FENO levels [74], has anti-inflammatory effects on small airways, has an influence on airway remodeling [75], and is effective in the treatment of chronic rhinosinusitis with nasal polyps in severe asthma [76]. Furthermore, other studies demonstrated a positive effect of biological drugs [77,78]. In our group, anti-IgE therapy led to a reduction in the need for systemic corticosteroids and a reduction in the number of exacerbations, as well as a gradual improvement in fitness and a return to physical activities, but the persistent overweight from the period before the initiation of anti-IgE therapy seems to be one of the possible reasons for the higher risk of CVD in some patients, in addition to the severity of the disease. The effect of LTRA therapy on improving control in patients with severe asthma on biological therapy was investigated by Quaranta et al. [79].

### Benefits and Limitations

The benefits of RHI measurement in the evaluation of vascular endothelial function and the prediction of CVD prognosis were confirmed in previous studies [80]. Using a new plethysmographic method for RHI evaluation has several advantages. Our study has relevant strengths, which are summarized as follows: non-invasiveness, high sensitivity, low biological variability, and the objectivity of the results due to validated automatic processing which substantially reduced the human factor to strengthen the study. Based on that, the simultaneous measurement of the contralateral non-occluded arm allowed the elimination of unintended exogenous changes during the examination. The close relationship between decreased RHI and coronary dysfunction was confirmed by several invasive methods [81]. Another strength is that Czech children, adolescents, and young adults with asthma represent the clinical and ethnic homogeneity of the type of patients who have been included in the study, which is a clear advantage in such studies. However, several limitations must be considered in interpreting the results of our study. As an observational study, despite an unequivocal association we could not prove causality; therefore, we stress that our results should be interpreted with caution. The primary outcome of this research was to evaluate the association between asthma in young subjects and CVD risk. Given the design of our study, we cannot clearly delineate the biological mechanisms that lead to a higher risk of CVD. Furthermore, we cannot exclude the possibility that the observed differences were related to ongoing medication. Further studies exploring the role of therapeutic interventions in relation to ED are needed to better understand the influence of therapeutic approaches on endothelial function. Although our data should be interpreted with caution, we believe that it retains substantial relevance, and the several potential limitations did not compromise our extrapolations. Nevertheless, if an association was established, it is important to make clear that it was from a hypothesis-generating study; consequently, future confirmatory studies are needed to clarify the strength of any association and disentangle the limitations of observational studies. Severe refractory asthma, especially the corticodependent type, may affect the tested biomarker, hsCRP, as well as adjunctive treatment of asthma with a macrolide antibiotic, but none of the asthmatics examined by us had this type of treatment. One of the limitations is the absence of the T2-low phenotype in asthmatics with typical corticoresistance and fixed airway obstruction without the possibility of specific biological treatment, in whom it would be particularly interesting to monitor CVD risk. Further research regarding the T2-low phenotype is therefore warranted. Other limitations also lie in the number of probands, in the absence of a precise cutoff limit for RHI and vascular biomarkers in children, and in the uniformity of sensors with no definition of the minimum finger thickness, which has already been extensively discussed [20,81,82]. To avoid inaccurate results caused by measuring younger children with fingers too small for sensors, we selected participants older than 14 years, and this was also due to the association of younger age with lower RHI but not lower FMD among adolescents, as described by Kelly et al. [82]. 

It would be premature to recommend changes in the management of young asthmatics to prevent ED and CVD based on this single report. Nonetheless, we hope that our findings highlight areas for further research and will motivate a further search to more fully characterize the excess risk of CVD.

## 6. Conclusions

Our study clearly indicates that the risk of CVD is strongly associated with young asthmatic subjects. Evidence on the associations of asthma with ED and CVD risk is attracting increasing medical attention due to the high prevalence of asthma and the health and economic consequences of CVD [58]. Our study suggested that persistent asthmatics have a higher CVD risk than non-asthmatics and that severe refractory asthmatics are at the highest risk of CVD, although the precise mechanism is unclear. In particular, this correlation, including the lack of conventional CVD risk factors in asthmatics, is of paramount importance in considering the management of patients with asthma. Our study clearly demonstrated further evidence of the crucial need to develop strategies to evaluate whether the attenuation of systemic inflammation can help to mitigate and prevent vascular inflammation and CVD risks associated with asthma. We should pay attention to these patients, as there is an urgent unmet need for further research to illuminate possible mechanisms and to determine whether there may be some advantage to be gained in selectively manipulating public health interventions in an attempt to substantially decrease this risk in high-risk young asthma patients. 

## Figures and Tables

**Figure 1 biomedicines-10-02614-f001:**
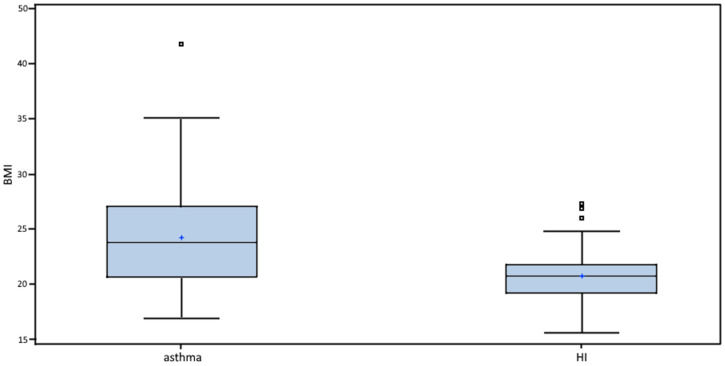
BMI in asthmatics vs. HI. Boxes indicate the inter-quartile range. Horizontal lines within boxes indicate medians. Whiskers extend to the highest or lowest values. BMI: body mass index; HI: healthy individuals.

**Figure 2 biomedicines-10-02614-f002:**
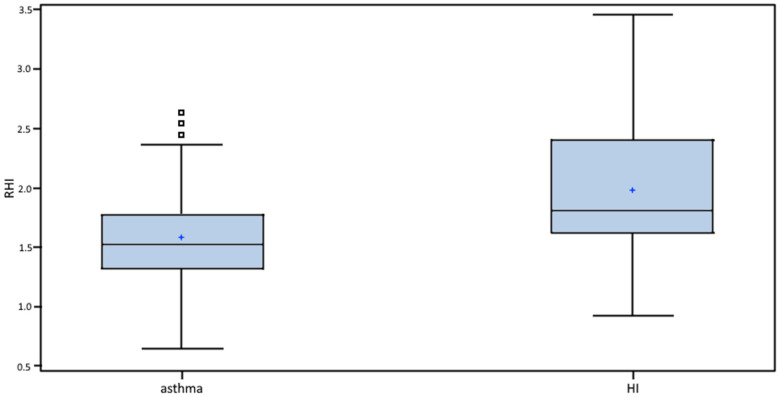
RHI in asthmatics vs. HI. Boxes indicate the inter-quartile range. Horizontal lines within boxes indicate medians. Whiskers extend to the highest or lowest values. RHI: reactive hyperemic index; HI: healthy individuals.

**Figure 3 biomedicines-10-02614-f003:**
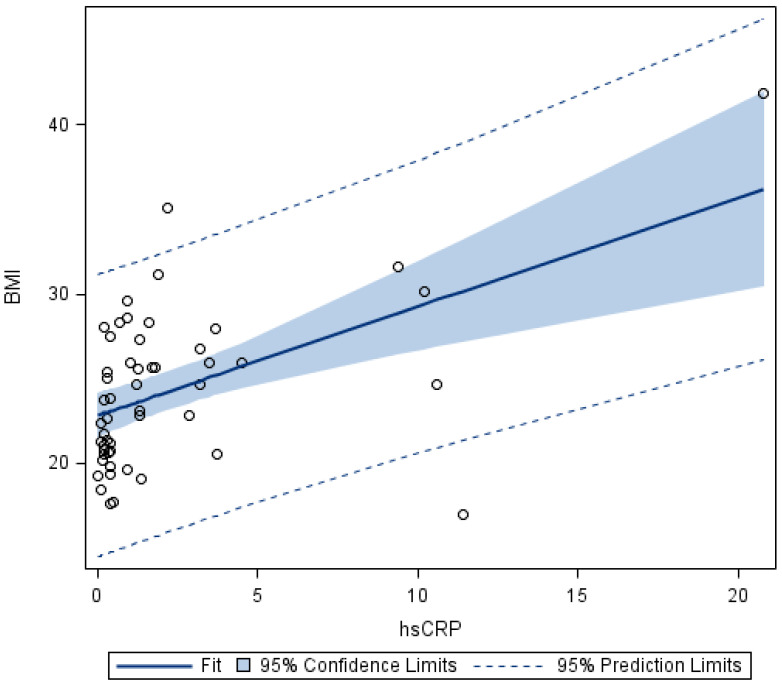
Linear regression of BMI and hsCRP in asthmatics. Moderately strong, statistically significant correlation confirmed. BMI: body mass index; hsCRP: high-sensitive C-reactive protein.

**Table 1 biomedicines-10-02614-t001:** Demographic data.

Data	Asthma Group Median (min; max)	Comparison Group Median (min; max)	Statistical Significance
Number	52	45	
Gender (male/female)	22/30	18/27	Equivalency (±20)
Age (y)	25.22 (13.97; 49.85)	17.04 (11.79; 28.32)	*p* < 0.005
BMI (kg/m^2^)	23.77 (16.96; 41.87)	20.73 (15.56; 27.31)	*p* < 0.001
Systolic blood pressure (mm Hg)	119 (98; 150)	116 (90; 136)	NS
Diastolic blood pressure (mm Hg)	74.5 (56; 95)	70 (50; 88)	*p* < 0.005

Values are expressed as the median with minimum and maximum values in parentheses; BMI: body mass index; NS: non-significant.

**Table 2 biomedicines-10-02614-t002:** Asthmatics: file properties.

Data	Number	%
Asthma type		
Allergic: persistent mild	28	53.85
Allergic: persistent moderate	8	15.38
Allergic: persistent severe and severe refractory	10	19.23
Exercise-induced	6	11.54
FENO (ppb)		
Normal (<25)	31	59.62
Mid-range (25–50)	10	19.23
Positive (>50)	11	21.15
Spirometry: peripheral airway obstruction		
Normal (FEF_25__–50_ > 66%)	33	63.46
Mild (FEF_25__–50_ 55–66%)	9	17.31
Moderate (FEF_25__–50_ 45–54%)	4	7.69
Severe and very severe (FEF_25__–50_ < 45%)	6	11.54
Asthma control test (score 5–25)		
Full-controlled asthma (score 25)	24	46.15
Well-controlled asthma (score 20–24)	12	23.08
Poor control of asthma (score < 20)	16	30.77

FENO: fractional exhaled nitric oxide, values in parts per billion (ppb). FEF_25–50_: the forced expiratory flow between 25 and 50% of the forced vital capacity.

**Table 3 biomedicines-10-02614-t003:** Asthma treatment according to asthma severity.

	Asthma Severity				
Type of Treatment	Mild	Moderate	Severe	Exercise Induced	Total
LTRA	4	0	0	0	4
LTRA + antiHIS	3	0	0	0	3
ICS	3	1	0	1	5
ICS + LTRA	2	0	0	0	2
ICS + antiHIS	2	0	0	0	2
ICS + antiHIS + LTRA	3	0	0	0	3
ICS + LABA	2	2	0	2	6
ICS + LABA + LTRA	0	0	1	0	1
ICS + LABA + antiHIS	5	3	0	3	11
ICS + LABA + antiHIS + LTRA	4	2	0	0	6
ICS + LABA + LTRA + antiHIS + OMA	0	0	2	0	2
ICS + LABA + LAMA + LTRA + OMA	0	0	1	0	1
ICS + LABA + OCS + OMA	0	0	1	0	1
ICS + LABA + LAMA + antiHIS + OCS + OMA	0	0	1	0	1
ICS + LABA + LAMA + LTRA + antiHIS + OCS + OMA	0	0	1	0	1
ICS + LABA + LAMA + LTRA + antiHIS + TPH + OMA	0	0	1	0	1
ICS + LABA + LAMA + antiHIS + TPH + OCS + OMA	0	0	2	0	2

LTRA: leukotriene receptor antagonist; antiHIS: antihistamines; ICS: inhaled corticosteroid; LABA: long-acting beta-agonist; LAMA: long-acting beta-agonist; OCS: oral corticosteroid; OMA: omalizumab; TPH: theophylline.

**Table 4 biomedicines-10-02614-t004:** Laboratory markers of allergy according to asthma severity.

Marker	Asthma Type	Positive Patients		Average (min; max)
		Count	% of all Positive	
Total IgE (positive >100 kIU/L)	Mild	15	47	421.5 (108.0; 1770.0)
	Moderate	7	22	705.5 (159.0; 2340.0)
	Severe	9	28	755.3 (178.0; 1060.0)
	Exercise-induced	1	3	110.0
Allergen-specific IgE (positive >0.35 U/mL)	Mild	25	54	
	Moderate	6	13	
	Severe	10	22	
	Exercise-induced	5	11	
ECP (positive >24 µg/L)	Mild	14	50	23.8 (24.0; 83.60)
	Moderate	3	11	97.2 (31.9; 200.0)
	Severe	9	32	78.0 (34.1; 200.0)
	Exercise-induced	2	7	55.9 (52.80; 59.0)
Blood Absolute Eosinophil Count (positive ≥0.3 × 10⁹/L)	Mild	11	50	0.46 (0.30; 1.00)
	Moderate	5	23	0.52 (0.30; 1.20)
	Severe	5	23	0.68 (0.30; 1.30)
	Exercise-induced	1	4	0.30

IgE: immunoglobulin E; ECP: eosinophil cationic protein.

**Table 5 biomedicines-10-02614-t005:** RHI and biomarker values.

Data	Asthma Group Median (min; max)	Comparison Group Median (min; max)	Statistical Significance
RHI	1.53 (0.65; 2.64)	1.81 (0.93; 3.46)	*p* < 0.001
hsCRP (mg/L)	0.90 (0.02; 20.80)	0.25 (0.02; 1.64)	*p* < 0.001
VCAM-1 (μg/L)	758.10 (403.1; 1414.9)	942.15 (555.5; 1518.4)	*p* < 0.01
E-selectin (μg/L)	53.00 (10.10; 141.80)	53.20 (18.70; 145.80)	NS
ADMA (μmol/L)	0.44 (0.25; 0.83)	0.54 (0.31; 0.89)	*p* < 0.05

Values are expressed as the median with minimum and maximum values in parentheses; RHI: reactive hyperemic index; hsCRP: high sensitive C-reactive protein; VCAM-1: vascular cell adhesive molecule; ADMA: asymmetric dimethylarginine; NS: non-significant.

## Data Availability

The data presented in this study are available upon request from the first author for privacy reasons.

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
