# Peer review of "Bronchial Asthma as a Cardiovascular Risk Factor: A Prospective Observational Study"

_biomedicines, 2022, doi:10.3390/biomedicines10102614_

Round 1
Reviewer 1 Report
The article is deals with the bronchial asthma as a cardiovascular risk factor for patients with bronchial asthma. It is very important topic as far as a great number of people suffer from asthma and from cardiovascular diseases.
I have some comments:
line 24, further in the text, in the tables and figures: “healthy control” should be replaced on “healthy individuals” or “comparison group” (instead of “control group”). You can't experiment on humans. Control group is used during the experiments on animals.
line 79: the word “an important” better to delete or replace on another word, for example “severe”
line 84: “NO gene polymorphism” - what does it mean? Perhaps it should be “nitric oxide synthase (NOS) gene polymorphism”?
The aim of the study was …..
lines 94-96: to evaluate relationships between RHI or biomarkers to associated CVD risk factors, such as nutritional status, pulmonary function, endobronchial eosinophilic inflammation, and control and severity of asthma.
lines 160-161: “All subjects or their legal representatives gave their informed consent for inclusion before they participated in the study”
Add please “and data processing” after “inclusion”:
“All subjects or their legal representatives gave their informed consent for inclusion and data processing before they participated in the study”
In the text neither in Materials and methods nor in the Results there is no information about evaluation of nutritional status and eosinophilic inflammation. The aim of the study should be corrected.
It is desirable to analyze or discuss does the kind of therapy influence on the tested biomarkers.
Reviewer 2 Report
I have read the original work with attention and interest. In various fields some points emerge.
- Methods section
Some crucial points arise that need to be investigated. The type of prospectively selected asthmatic patients is not clear. From what has been written, these are asthmatic patients with asthma of varying severity. Table 2 considers allergic persisten mild, allergic persistent moderate, allergic persistent severe and severe refractory, and non allergic (exrcise induced). Patients with severe persistent eosinophilic asthma (Th2 high or low?) Are therefore not considered. In the most recent current classifications, patients with severe asthma are often phenotyped precisely in relation to the Th2 high or low phenotype. In fact, this subdivision also allows to evaluate the choice of the biological drug to be used for the therapy, also in relation to the number of eosinophils. From what is written some patients were on therapy with biological drugs (monoclonal antibodies) only with anti-IgE. Therefore it would be better to clarify whether it is only allergic patients receiving omalizumab (anti-IgE) and specify it in the table. Therefore, it would also be useful to clarify that the asthmatic patients considered belong only to an essentially allergic phenotype with or without eosinophilia. It would also be useful to understand the level of IgE and eosinophils of the patients and to insert these data in the table.
2) How are patients with refractory asthma (therefore severe asthma) considered? Are they patients who are always poorly controlled despite maximal and anti-IgE tearpia? It would be useful to explain the reason for this lack of clinical response. Is their increased cardiovascular risk attributable to poor adherence to therapy which results in a lack of asthma control?
3) In relation to what is written by the authors, these are patients with variable asthma drugs. In the table it would be useful to specify what kind of therapy they take and not describe it only in the results. This would make it clearer to the reader who the patient is. Furthermore, as a guide, patients with severe (refractory) asthma, patients should be on maximal therapy or be treated with ICS / LABA + LAMA. The authors, on the other hand, remain vague and it is not clear in relation to the severity of asthma what type of therapy they take. Furthermore, approximately "13.5% of patients were only on leukotriene receptor antagonist treatment and 11.5% of examined asthmatics had severe or very severe peripheral airway obstruction on 182 regular therapy". Which group of asthma patients do they belong to? Were the first group non-allergic patients (13.5%)? The second group (11.5) what treatment do they take for the serious small airway obstruction and what group do they belong to? (mild, moderate or severe?).
4) There are works in the literature that highlight the clinical response also in relation to the treatment with current biological drugs in relation to the present comorbidities that the authors do not mention, but that can be included in the introduction or in the discussion. For example:
a) https://doi.org/10.1016/j.waojou.2020.100103
b) DOI: 10.1016 / j.rmed.2021.106491
- Discussion section
It would be necessary to deepen or insert a paragraph on therapies and clinical response. Response to therapies can also improve cardiovascular outcomes.
For the rest, the discussion is well written and structured.
- Improve native english
- Major spelling check.
Round 2
Reviewer 2 Report
I have carefully evaluated the changes made by the authors in relation to the indications of the reviewers and I believe that it can be accepted for publication